# Glow: Generative Flow with Invertible $1\times1$ Convolutions

**Diederik P. Kingma**[*][†], **Prafulla Dhariwal**[*]
[*]OpenAI
[†]Google AI

## Abstract

Flow-based generative models (Dinh et al., 2014) are conceptually attractive due to tractability of the exact log-likelihood, tractability of exact latent-variable inference, and parallelizability of both training and synthesis. In this paper we propose *Glow*, a simple type of generative flow using an invertible $1 \times 1$ convolution. Using our method we demonstrate a significant improvement in log-likelihood on standard benchmarks. Perhaps most strikingly, we demonstrate that a flow-based generative model optimized towards the plain log-likelihood objective is capable of efficient realistic-looking synthesis and manipulation of large images. The code for our model is available at `https://github.com/openai/glow`.

## 1 Introduction

Two major unsolved problems in the field of machine learning are (1) data-efficiency: the ability to learn from few datapoints, like humans; and (2) generalization: robustness to changes of the task or its context. AI systems, for example, often do not work at all when given inputs that are different

---

[*]Equal contribution.

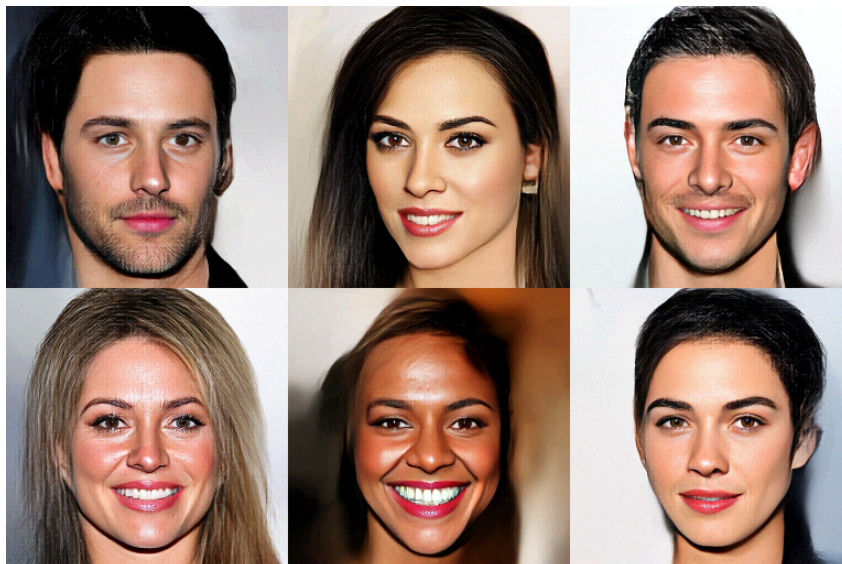

Figure 1: Synthetic celebrities sampled from our model; see Section 3 for architecture and method, and Section 5 for more results.

from their training distribution. A promise of *generative models*, a major branch of machine learning, is to overcome these limitations by: (1) learning realistic world models, potentially allowing agents to plan in a world model before actual interaction with the world, and (2) learning meaningful features of the input while requiring little or no human supervision or labeling. Since such features can be learned from large unlabeled datasets and are not necessarily task-specific, downstream solutions based on those features could potentially be more robust and more data efficient. In this paper we work towards this ultimate vision, in addition to intermediate applications, by aiming to improve upon the state-of-the-art of generative models.

Generative modeling is generally concerned with the extremely challenging task of modeling all dependencies within very high-dimensional input data, usually specified in the form of a full joint probability distribution. Since such joint models potentially capture all patterns that are present in the data, the applications of accurate generative models are near endless. Immediate applications are as diverse as speech synthesis, text analysis, semi-supervised learning and model-based control; see Section 4 for references.

The discipline of generative modeling has experienced enormous leaps in capabilities in recent years, mostly with likelihood-based methods (Graves, 2013; Kingma and Welling, 2013, 2018; Dinh et al., 2014; van den Oord et al., 2016a) and generative adversarial networks (GANs) (Goodfellow et al., 2014) (see Section 4). Likelihood-based methods can be divided into three categories:

1. Autoregressive models (Hochreiter and Schmidhuber, 1997; Graves, 2013; van den Oord et al., 2016a,b; Van Den Oord et al., 2016). Those have the advantage of simplicity, but have as disadvantage that synthesis has limited parallelizability, since the computational length of synthesis is proportional to the dimensionality of the data; this is especially troublesome for large images or video.

2. Variational autoencoders (VAEs) (Kingma and Welling, 2013, 2018), which optimize a lower bound on the log-likelihood of the data. Variational autoencoders have the advantage of parallelizability of training and synthesis, but can be comparatively challenging to optimize (Kingma et al., 2016).

3. Flow-based generative models, first described in NICE (Dinh et al., 2014) and extended in RealNVP (Dinh et al., 2016). We explain the key ideas behind this class of model in the following sections.

Flow-based generative models have so far gained little attention in the research community compared to GANs (Goodfellow et al., 2014) and VAEs (Kingma and Welling, 2013). Some of the merits of flow-based generative models include:

- Exact latent-variable inference and log-likelihood evaluation. In VAEs, one is able to infer only approximately the value of the latent variables that correspond to a datapoint. GAN's have no encoder at all to infer the latents. In reversible generative models, this can be done exactly without approximation. Not only does this lead to accurate inference, it also enables optimization of the exact log-likelihood of the data, instead of a lower bound of it.

- Efficient inference and efficient synthesis. Autoregressive models, such as the Pixel-CNN (van den Oord et al., 2016b), are also reversible, however synthesis from such models is difficult to parallelize, and typically inefficient on parallel hardware. Flow-based generative models like Glow (and RealNVP) are efficient to parallelize for both inference and synthesis.

- Useful latent space for downstream tasks. The hidden layers of autoregressive models have unknown marginal distributions, making it much more difficult to perform valid manipulation of data. In GANs, datapoints can usually not be directly represented in a latent space, as they have no encoder and might not have full support over the data distribution. (Grover et al., 2018). This is not the case for reversible generative models and VAEs, which allow for various applications such as interpolations between datapoints and meaningful modifications of existing datapoints.

- Significant potential for memory savings. Computing gradients in reversible neural networks requires an amount of memory that is constant instead of linear in their depth, as explained in the RevNet paper (Gomez et al., 2017).

In this paper we propose a new a generative flow coined *Glow*, with various new elements as described in Section 3. In Section 5, we compare our model quantitatively with previous flows, and in Section 6, we study the qualitative aspects of our model on high-resolution datasets.

## 2 Background: Flow-based Generative Models

Let $\mathbf{x}$ be a high-dimensional random vector with unknown true distribution $\mathbf{x} \sim p^*(\mathbf{x})$. We collect an i.i.d. dataset $\mathcal{D}$, and choose a model $p_{\boldsymbol{\theta}}(\mathbf{x})$ with parameters $\boldsymbol{\theta}$. In case of discrete data $\mathbf{x}$, the log-likelihood objective is then equivalent to minimizing:

$$\mathcal{L}(\mathcal{D}) = \frac{1}{N} \sum_{i=1}^{N} - \log p_{\boldsymbol{\theta}}(\mathbf{x}^{(i)}) \tag{1}$$

In case of *continuous* data $\mathbf{x}$, we minimize the following:

$$\mathcal{L}(\mathcal{D}) \simeq \frac{1}{N} \sum_{i=1}^{N} - \log p_{\boldsymbol{\theta}}(\tilde{\mathbf{x}}^{(i)}) + c \tag{2}$$

where $\tilde{\mathbf{x}}^{(i)} = \mathbf{x}^{(i)} + u$ with $u \sim \mathcal{U}(0, a)$, and $c = -M \cdot \log a$ where $a$ is determined by the discretization level of the data and $M$ is the dimensionality of $\mathbf{x}$. Both objectives (eqs. (1) and (2)) measure the expected compression cost in nats or bits; see (Dinh et al., 2016). Optimization is done through stochastic gradient descent using minibatches of data (Kingma and Ba, 2015).

In most flow-based generative models (Dinh et al., 2014, 2016), the generative process is defined as:

$$\mathbf{z} \sim p_{\boldsymbol{\theta}}(\mathbf{z}) \tag{3}$$
$$\mathbf{x} = \mathbf{g}_{\boldsymbol{\theta}}(\mathbf{z}) \tag{4}$$

where $\mathbf{z}$ is the latent variable and $p_{\boldsymbol{\theta}}(\mathbf{z})$ has a (typically simple) tractable density, such as a spherical multivariate Gaussian distribution: $p_{\boldsymbol{\theta}}(\mathbf{z}) = \mathcal{N}(\mathbf{z}; 0, \mathbf{I})$. The function $\mathbf{g}_{\boldsymbol{\theta}}(..)$ is invertible, also called *bijective*, such that given a datapoint $\mathbf{x}$, latent-variable inference is done by $\mathbf{z} = \mathbf{f}_{\boldsymbol{\theta}}(\mathbf{x}) = \mathbf{g}_{\boldsymbol{\theta}}^{-1}(\mathbf{x})$. For brevity, we will omit subscript $\boldsymbol{\theta}$ from $\mathbf{f}_{\boldsymbol{\theta}}$ and $\mathbf{g}_{\boldsymbol{\theta}}$.

We focus on functions where $\mathbf{f}$ (and, likewise, $\mathbf{g}$) is composed of a sequence of transformations: $\mathbf{f} = \mathbf{f}_1 \circ \mathbf{f}_2 \circ \cdots \circ \mathbf{f}_K$, such that the relationship between $\mathbf{x}$ and $\mathbf{z}$ can be written as:

$$\mathbf{x} \xleftrightarrow{\mathbf{f}_1} \mathbf{h}_1 \xleftrightarrow{\mathbf{f}_2} \mathbf{h}_2 \cdots \xleftrightarrow{\mathbf{f}_K} \mathbf{z} \tag{5}$$

Such a sequence of invertible transformations is also called a (normalizing) *flow* (Rezende and Mohamed, 2015). Under the *change of variables* of eq. (4), the probability density function (pdf) of the model given a datapoint can be written as:

$$\log p_{\boldsymbol{\theta}}(\mathbf{x}) = \log p_{\boldsymbol{\theta}}(\mathbf{z}) + \log |\det(d\mathbf{z}/d\mathbf{x})| \tag{6}$$

$$= \log p_{\boldsymbol{\theta}}(\mathbf{z}) + \sum_{i=1}^{K} \log |\det(d\mathbf{h}_i/d\mathbf{h}_{i-1})| \tag{7}$$

where we define $\mathbf{h}_0 \triangleq \mathbf{x}$ and $\mathbf{h}_K \triangleq \mathbf{z}$ for conciseness. The scalar value $\log |\det(d\mathbf{h}_i/d\mathbf{h}_{i-1})|$ is the logarithm of the absolute value of the determinant of the Jacobian matrix $(d\mathbf{h}_i/d\mathbf{h}_{i-1})$, also called the *log-determinant*. This value is the change in log-density when going from $\mathbf{h}_{i-1}$ to $\mathbf{h}_i$ under transformation $\mathbf{f}_i$. While it may look intimidating, its value can be surprisingly simple to compute for certain choices of transformations, as previously explored in (Deco and Brauer, 1995; Dinh et al., 2014; Rezende and Mohamed, 2015; Kingma et al., 2016). The basic idea is to choose transformations whose Jacobian $d\mathbf{h}_i/d\mathbf{h}_{i-1}$ is a triangular matrix. For those transformations, the log-determinant is simple:

$$\log |\det(d\mathbf{h}_i/d\mathbf{h}_{i-1})| = \mathtt{sum}(\log |\mathtt{diag}(d\mathbf{h}_i/d\mathbf{h}_{i-1})|) \tag{8}$$

where $\mathtt{sum}()$ takes the sum over all vector elements, $\log()$ takes the element-wise logarithm, and $\mathtt{diag}()$ takes the diagonal of the Jacobian matrix.

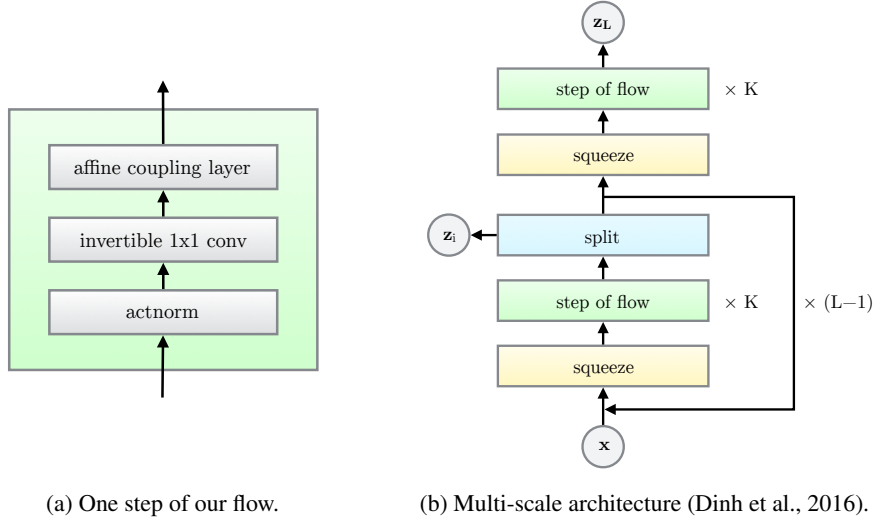

(a) One step of our flow.    (b) Multi-scale architecture (Dinh et al., 2016).

Figure 2: We propose a generative flow where each step (left) consists of an *actnorm* step, followed by an invertible $1 \times 1$ convolution, followed by an affine transformation (Dinh et al., 2014). This flow is combined with a multi-scale architecture (right). See Section 3 and Table 1.

Table 1: The three main components of our proposed flow, their reverses, and their log-determinants. Here, $\mathbf{x}$ signifies the input of the layer, and $\mathbf{y}$ signifies its output. Both $\mathbf{x}$ and $\mathbf{y}$ are tensors of shape $[h \times w \times c]$ with spatial dimensions $(h, w)$ and channel dimension $c$. With $(i, j)$ we denote spatial indices into tensors $\mathbf{x}$ and $\mathbf{y}$. The function $\texttt{NN}()$ is a nonlinear mapping, such as a (shallow) convolutional neural network like in ResNets (He et al., 2016) and RealNVP (Dinh et al., 2016).

| Description | Function | Reverse Function | Log-determinant |
|---|---|---|---|
| Actnorm. See Section 3.1. | $\forall i, j : \mathbf{y}_{i,j} = \mathbf{s} \odot \mathbf{x}_{i,j} + \mathbf{b}$ | $\forall i, j : \mathbf{x}_{i,j} = (\mathbf{y}_{i,j} - \mathbf{b})/\mathbf{s}$ | $h \cdot w \cdot \texttt{sum}(\log|\mathbf{s}|)$ |
| Invertible $1 \times 1$ convolution. $\mathbf{W} : [c \times c]$. See Section 3.2. | $\forall i, j : \mathbf{y}_{i,j} = \mathbf{W}\mathbf{x}_{i,j}$ | $\forall i, j : \mathbf{x}_{i,j} = \mathbf{W}^{-1}\mathbf{y}_{i,j}$ | $h \cdot w \cdot \log|\det(\mathbf{W})|$ or $h \cdot w \cdot \texttt{sum}(\log|\mathbf{s}|)$ (see eq. (10)) |
| Affine coupling layer. See Section 3.3 and (Dinh et al., 2014) | $\mathbf{x}_a, \mathbf{x}_b = \texttt{split}(\mathbf{x})$ $(\log \mathbf{s}, \mathbf{t}) = \texttt{NN}(\mathbf{x}_b)$ $\mathbf{s} = \exp(\log \mathbf{s})$ $\mathbf{y}_a = \mathbf{s} \odot \mathbf{x}_a + \mathbf{t}$ $\mathbf{y}_b = \mathbf{x}_b$ $\mathbf{y} = \texttt{concat}(\mathbf{y}_a, \mathbf{y}_b)$ | $\mathbf{y}_a, \mathbf{y}_b = \texttt{split}(\mathbf{y})$ $(\log \mathbf{s}, \mathbf{t}) = \texttt{NN}(\mathbf{y}_b)$ $\mathbf{s} = \exp(\log \mathbf{s})$ $\mathbf{x}_a = (\mathbf{y}_a - \mathbf{t})/\mathbf{s}$ $\mathbf{x}_b = \mathbf{y}_b$ $\mathbf{x} = \texttt{concat}(\mathbf{x}_a, \mathbf{x}_b)$ | $\texttt{sum}(\log(|\mathbf{s}|))$ |

# 3 Proposed Generative Flow

We propose a new flow, building on the NICE and RealNVP flows proposed in (Dinh et al., 2014, 2016). It consists of a series of steps of flow, combined in a multi-scale architecture; see Figure 2. Each step of flow consists of *actnorm* (Section 3.1) followed by an *invertible $1 \times 1$ convolution* (Section 3.2), followed by a coupling layer (Section 3.3).

This flow is combined with a multi-scale architecture; due to space constraints we refer to (Dinh et al., 2016) for more details. This architecture has a depth of flow $K$, and number of levels $L$ (Figure 2).

## 3.1 Actnorm: scale and bias layer with data dependent initialization

In Dinh et al. (2016), the authors propose the use of batch normalization (Ioffe and Szegedy, 2015) to alleviate the problems encountered when training deep models. However, since the variance of

activations noise added by batch normalization is inversely proportional to minibatch size per GPU or other processing unit (PU), performance is known to degrade for small per-PU minibatch size. For large images, due to memory constraints, we learn with minibatch size 1 per PU. We propose an *actnorm* layer (for *activation normalizaton*), that performs an affine transformation of the activations using a scale and bias parameter per channel, similar to batch normalization. These parameters are initialized such that the post-actnorm activations per-channel have zero mean and unit variance given an initial minibatch of data. This is a form of *data dependent initialization* (Salimans and Kingma, 2016). After initialization, the scale and bias are treated as regular trainable parameters that are independent of the data.

## 3.2 Invertible $1 \times 1$ convolution

(Dinh et al., 2014, 2016) proposed a flow containing the equivalent of a permutation that reverses the ordering of the channels. We propose to replace this fixed permutation with a (learned) invertible $1 \times 1$ convolution, where the weight matrix is initialized as a random rotation matrix. Note that a $1 \times 1$ convolution with equal number of input and output channels is a generalization of a permutation operation.

The log-determinant of an invertible $1 \times 1$ convolution of a $h \times w \times c$ tensor $\mathbf{h}$ with $c \times c$ weight matrix $\mathbf{W}$ is straightforward to compute:

$$\log \left| \det \left( \frac{d\, \texttt{conv2D}(\mathbf{h}; \mathbf{W})}{d\, \mathbf{h}} \right) \right| = h \cdot w \cdot \log |\det(\mathbf{W})| \tag{9}$$

The cost of computing or differentiating $\det(\mathbf{W})$ is $\mathcal{O}(c^3)$, which is often comparable to the cost computing $\texttt{conv2D}(\mathbf{h}; \mathbf{W})$ which is $\mathcal{O}(h \cdot w \cdot c^2)$. We initialize the weights $\mathbf{W}$ as a random rotation matrix, having a log-determinant of 0; after one SGD step these values start to diverge from 0.

**LU Decomposition.** This cost of computing $\det(\mathbf{W})$ can be reduced from $\mathcal{O}(c^3)$ to $\mathcal{O}(c)$ by parameterizing $\mathbf{W}$ directly in its LU decomposition:

$$\mathbf{W} = \mathbf{P}\mathbf{L}(\mathbf{U} + \text{diag}(\mathbf{s})) \tag{10}$$

where $\mathbf{P}$ is a permutation matrix, $\mathbf{L}$ is a lower triangular matrix with ones on the diagonal, $\mathbf{U}$ is an upper triangular matrix with zeros on the diagonal, and $\mathbf{s}$ is a vector. The log-determinant is then simply:

$$\log |\det(\mathbf{W})| = \texttt{sum}(\log |\mathbf{s}|) \tag{11}$$

The difference in computational cost will become significant for large $c$, although for the networks in our experiments we did not measure a large difference in wallclock computation time.

In this parameterization, we initialize the parameters by first sampling a random rotation matrix $\mathbf{W}$, then computing the corresponding value of $\mathbf{P}$ (which remains fixed) and the corresponding initial values of $\mathbf{L}$ and $\mathbf{U}$ and $\mathbf{s}$ (which are optimized).

## 3.3 Affine Coupling Layers

A powerful reversible transformation where the forward function, the reverse function and the log-determinant are computationally efficient, is the *affine coupling* layer introduced in (Dinh et al., 2014, 2016). See Table 1. An *additive coupling layer* is a special case with $\mathbf{s} = 1$ and a log-determinant of 0.

**Zero initialization.** We initialize the last convolution of each $\texttt{NN}()$ with zeros, such that each affine coupling layer initially performs an identity function; we found that this helps training very deep networks.

**Split and concatenation.** As in (Dinh et al., 2014), the $\texttt{split}()$ function splits $\mathbf{h}$ the input tensor into two halves along the channel dimension, while the $\texttt{concat}()$ operation performs the corresponding reverse operation: concatenation into a single tensor. In (Dinh et al., 2016), another type of split was introduced: along the spatial dimensions using a checkerboard pattern. In this work we only perform splits along the channel dimension, simplifying the overall architecture.

**Permutation.** Each step of flow above should be preceded by some kind of permutation of the variables that ensures that after sufficient steps of flow, each dimensions can affect every other dimension. The type of permutation specifically done in (Dinh et al., 2014, 2016) is equivalent to simply *reversing the ordering* of the channels (features) before performing an additive coupling layer. An alternative is to perform a (fixed) random permutation. Our invertible 1x1 convolution is a generalization of such permutations. In experiments we compare these three choices.

# 4   Related Work

This work builds upon the ideas and flows proposed in (Dinh et al., 2014) (NICE) and (Dinh et al., 2016) (RealNVP); comparisons with this work are made throughout this paper. In (Papamakarios et al., 2017) (MAF), the authors propose a generative flow based on IAF (Kingma et al., 2016); however, since synthesis from MAF is non-parallelizable and therefore inefficient, we omit it from comparisons. Synthesis from autoregressive (AR) models (Hochreiter and Schmidhuber, 1997; Graves, 2013; van den Oord et al., 2016a,b; Van Den Oord et al., 2016) is similarly non-parallelizable. Synthesis of high-dimensional data typically takes multiple orders of magnitude longer with AR models; see (Kingma et al., 2016; Oord et al., 2017) for evidence. Sampling $256 \times 256$ images with our largest models takes less than one second on current hardware. [2] (Reed et al., 2017) explores techniques for speeding up synthesis in AR models considerably; we leave the comparison to this line of work to future work.

GANs (Goodfellow et al., 2014) are arguably best known for their ability to synthesize large and realistic images (Karras et al., 2017), in contrast with likelihood-based methods. Downsides of GANs are their general lack of latent-space encoders, their general lack of full support over the data (Grover et al., 2018), their difficulty of optimization, and their difficulty of assessing overfitting and generalization.

# 5   Quantitative Experiments

We begin our experiments by comparing how our new flow compares against RealNVP (Dinh et al., 2016). We then apply our model on other standard datasets and compare log-likelihoods against previous generative models. See the appendix for optimization details. In our experiments, we let each `NN()` have three convolutional layers, where the two hidden layers have ReLU activation functions and 512 channels. The first and last convolutions are $3 \times 3$, while the center convolution is $1 \times 1$, since both its input and output have a large number of channels, in contrast with the first and last convolution.

**Gains using invertible $1 \times 1$ Convolution.** We choose the architecture described in Section 3, and consider three variations for the permutation of the channel variables - a reversing operation as described in the RealNVP, a fixed random permutation, and our invertible $1 \times 1$ convolution. We compare for models with only additive coupling layers, and models with affine coupling. As described earlier, we initialize all models with a data-dependent initialization which normalizes the activations of each layer. All models were trained with $K = 32$ and $L = 3$. The model with $1 \times 1$ convolution has a negligible $0.2\%$ larger amount of parameters.

We compare the average negative log-likelihood (bits per dimension) on the CIFAR-10 (Krizhevsky, 2009) dataset, keeping all training conditions constant and averaging across three random seeds. The results are in Figure 3. As we see, for both additive and affine couplings, the invertible $1 \times 1$ convolution achieves a lower negative log likelihood and converges faster. The affine coupling models also converge faster than the additive coupling models. We noted that the increase in wallclock time for the invertible $1 \times 1$ convolution model was only $\approx 7\%$, thus the operation is computationally efficient as well.

**Comparison with RealNVP on standard benchmarks.** Besides the permutation operation, the RealNVP architecture has other differences such as the spatial coupling layers. In order to verify that our proposed architecture is overall competitive with the RealNVP architecture, we compare

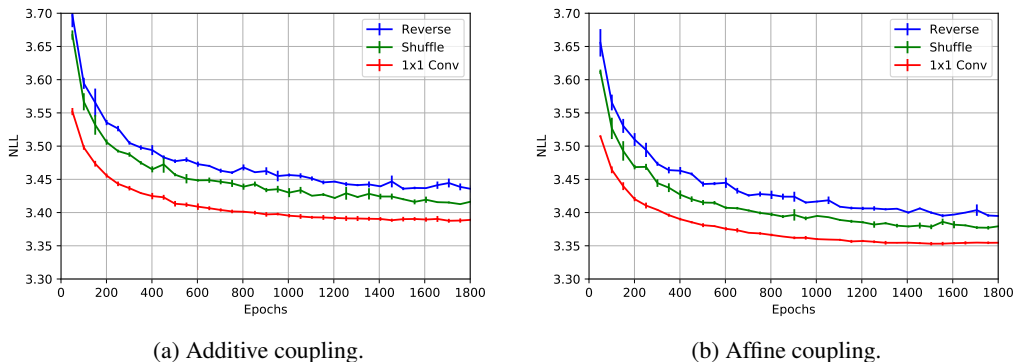

|                  |                  |
|:----------------:|:----------------:|
| (a) Additive coupling. | (b) Affine coupling. |

Figure 3: Comparison of the three variants - a reversing operation as described in the RealNVP, a fixed random permutation, and our proposed invertible $1 \times 1$ convolution, with additive (left) versus affine (right) coupling layers. We plot the mean and standard deviation across three runs with different random seeds.

Table 2: Best results in bits per dimension of our model compared to RealNVP.

| Model | CIFAR-10 | ImageNet 32x32 | ImageNet 64x64 | LSUN (bedroom) | LSUN (tower) | LSUN (church outdoor) |
|-------|----------|----------------|----------------|----------------|--------------|-----------------------|
| RealNVP | 3.49 | 4.28 | 3.98 | 2.72 | 2.81 | 3.08 |
| Glow | **3.35** | **4.09** | **3.81** | **2.38** | **2.46** | **2.67** |

our models on various natural images datasets. In particular, we compare on CIFAR-10, ImageNet (Russakovsky et al., 2015) and LSUN (Yu et al., 2015) datasets. We follow the same preprocessing as in (Dinh et al., 2016). For Imagenet, we use the $32 \times 32$ and $64 \times 64$ downsampled version of ImageNet (Oord et al., 2016), and for LSUN we downsample to $96 \times 96$ and take random crops of $64 \times 64$. We also include the bits/dimension for our model trained on $256 \times 256$ CelebA HQ used in our qualitative experiments.[3] As we see in Table 2, our model achieves a significant improvement on all the datasets.

## 6 Qualitative Experiments

We now study the qualitative aspects of the model on high-resolution datasets. We choose the CelebA-HQ dataset (Karras et al., 2017), which consists of 30000 high resolution images from the CelebA dataset, and train the same architecture as above but now for images at a resolution of $256^2$, $K = 32$ and $L = 6$. To improve visual quality at the cost of slight decrease in color fidelity, we train our models on 5-bit images. We aim to study if our model can scale to high resolutions, produce realistic samples, and produce a meaningful latent space. Due to device memory constraints, at these resolutions we work with minibatch size 1 per PU, and use gradient checkpointing (Salimans and Bulatov, 2017). In the future, we could use a constant amount of memory independent of depth by utilizing the reversibility of the model (Gomez et al., 2017).

Consistent with earlier work on likelihood-based generative models, we found that sampling from a reduced-temperature model (Parmar et al., 2018) often results in higher-quality samples. When sampling with temperature $T$, we sample from the distribution $p_{\boldsymbol{\theta}, T}(\mathbf{x}) \propto (p_{\boldsymbol{\theta}}(\mathbf{x}))^{T^2}$. In case of additive coupling layers, this can be achieved simply by multiplying the standard deviation of $p_{\boldsymbol{\theta}}(\mathbf{z})$ by a factor of $T$.

**Synthesis and Interpolation.** Figure 4 shows the random samples obtained from our model. The images are of high quality for a non-autoregressive likelihood based model. To see how well we can interpolate, we take a pair of real images, encode them with the encoder, and linearly interpolate

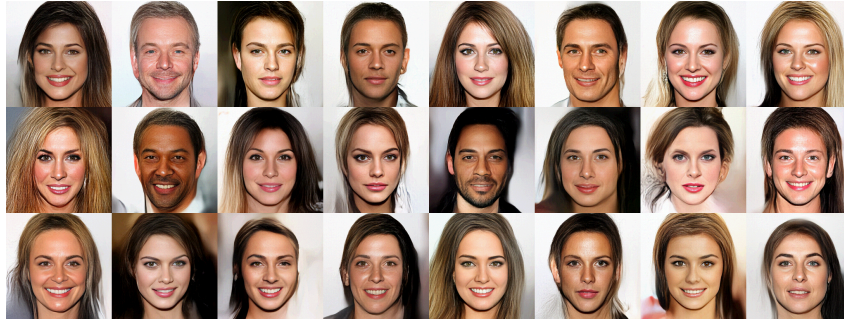

Figure 4: Random samples from the model, with temperature $0.7$.

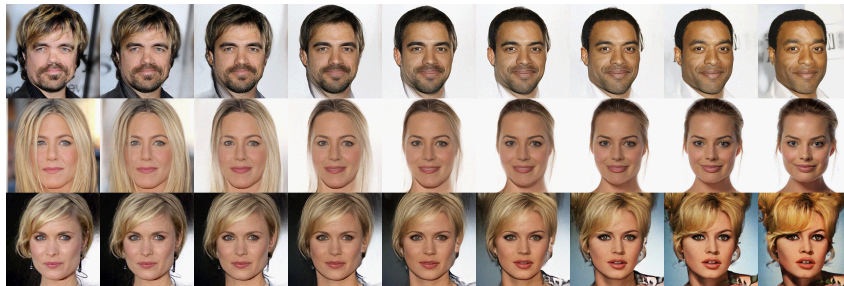

Figure 5: Linear interpolation in latent space between real images.

between the latents to obtain samples. The results in Figure 5 show that the image manifold of the generator distribution is smooth and almost all intermediate samples look like realistic faces.

**Semantic Manipulation.** We now consider modifying attributes of an image. To do so, we use the labels in the CelebA dataset. Each image has a binary label corresponding to presence or absence of attributes like smiling, blond hair, young, etc. This gives us 30000 binary labels for each attribute. We then calculate the average latent vector $\mathbf{z}_{pos}$ for images with the attribute and $\mathbf{z}_{neg}$ for images without, and then use the difference $(\mathbf{z}_{pos} - \mathbf{z}_{neg})$ as a direction for manipulating. Note that this is a relatively small amount of supervision, and is done after the model is trained (no labels were used while training), making it extremely easy to do for a variety of different target attributes. The results are shown in Figure 6 (appendix).

**Effect of temperature and model depth.** Figure 8 (appendix) shows how the sample quality and diversity varies with temperature. The highest temperatures have noisy images, possibly due to overestimating the entropy of the data distribution; we choose a temperature of $0.7$ as a sweet spot for diversity and quality of samples. Figure 9 (appendix) shows how model depth affects the ability of the model to learn long-range dependencies.

## 7    Conclusion

We propose a new type of generative flow and demonstrate improved quantitative performance in terms of log-likelihood on standard image modeling benchmarks. In addition, we demonstrate that when trained on high-resolution faces, our model is able to synthesize realistic images.

## Footnotes

[2]More specifically, generating a $256 \times 256$ image at batch size 1 takes about 130ms on a single NVIDIA GTX 1080 Ti, and about 550ms on a NVIDIA Tesla K80.

[3]Since the original CelebA HQ dataset didn't have a validation set, we separated it into a training set of 27000 images and a validation set of 3000 images.

[3]For $128 \times 128$ and $96 \times 96$ versions, we centre cropped the original image, and downsampled. For $64 \times 64$ version, we took random crops from the $96 \times 96$ downsampled image as done in Dinh et al. (2016)

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
