[Supplementary Material]

# A Figures

Figures 6, 8 and 9 are referenced from the main paper.

(a) Smiling

(b) Pale Skin

(c) Blond Hair

(d) Narrow Eyes

(e) Young

(f) Male

Figure 6: Manipulation of attributes of a face. Each row is made by interpolating the latent code of an image along a vector corresponding to the attribute, with the middle image being the original image.

Figure 7: Samples from model trained on 5-bit LSUN bedrooms, at temperature 0.875. Resolutions 64, 96 and 128 respectively [4].

Figure 8: Effect of change of temperature. From left to right, samples obtained at temperatures $0, 0.25, 0.6, 0.7, 0.8, 0.9, 1.0$.

Figure 9: Samples from shallow model on left vs deep model on right. Shallow model has $L = 4$ levels, while deep model has $L = 6$ levels.

## B  Additional quantitative results

See Table 3.

Table 3: Quantiative results in bits per dimension on the test set.

| Dataset | Glow |
|---|---|
| CIFAR-10, 32×32, 5-bit | 1.67 |
| ImageNet, 32×32, 5-bit | 1.99 |
| ImageNet, 64×64, 5-bit | 1.76 |
| CelebA HQ, 256×256, 5-bit | 1.03 |

## C  Simple python implementation of the invertible $1 \times 1$ convolution

```python
# Invertible 1x1 conv
def invertible_1x1_conv(z, logdet, forward=True):
    # Shape
    height, width, channels = z.shape[1:]

    # Sample a random orthogonal matrix to initialise weights
    w_init = np.linalg.qr(np.random.randn(channels,channels))[0]
    w = tf.get_variable("W", initializer=w_init)

    # Compute log determinant
    dlogdet = height * width * tf.log(abs(tf.matrix_determinant(w)))

    if forward:
        # Forward computation
        _w = tf.reshape(w, [1,1,channels,channels])
        z = tf.nn.conv2d(z, _w, [1,1,1,1], 'SAME')
        logdet += dlogdet

        return z, logdet
    else:
        # Reverse computation
        _w = tf.matrix_inverse(w)
        _w = tf.reshape(_w, [1,1,channels,channels])
        z = tf.nn.conv2d(z, _w, [1,1,1,1], 'SAME')
        logdet -= dlogdet

        return z, logdet
```

## D  Optimization details

We use the Adam optimizer (Kingma and Ba, 2015) with $\alpha = 0.001$ and default $\beta_1$ and $\beta_2$. In out quantitative experiments (Section 5, Table 2) we used the following hyperparameters (Table 4).

In our qualitative experiments (Section 6), we used the following hyperparameters (Table 5)

## E  Extra samples from qualitative experiments

For the class conditional CIFAR-10 and 32×32 ImageNet samples, we used the same hyperparameters as the quantitative experiments, but with a class dependent prior at the top-most level. We also added

Table 4: Hyperparameters for results in Section 5, Table 2.

| Dataset | Minibatch Size | Levels (L) | Depth per level (K) | Coupling |
|---|---|---|---|---|
| CIFAR-10 | 512 | 3 | 32 | Affine |
| ImageNet, 32×32 | 512 | 3 | 48 | Affine |
| ImageNet, 64×64 | 128 | 4 | 48 | Affine |
| LSUN, 64×64 | 128 | 4 | 48 | Affine |

Table 5: Hyperparameters for results in Section 6.

| Dataset | Minibatch Size | Levels (L) | Depth per level (K) | Coupling |
|---|---|---|---|---|
| LSUN, 64×64, 5-bit | 128 | 4 | 48 | Additive |
| LSUN, 96×96, 5-bit | 320 | 5 | 64 | Additive |
| LSUN, 128×128, 5-bit | 160 | 5 | 64 | Additive |
| CelebA HQ, 256×256, 5-bit | 40 | 6 | 32 | Additive |

a classification loss to predict the class label from the second last layer of the encoder, with a weight of $\lambda = 0.01$. The results are in Figure 10.

# F   Extra samples from the quantitative experiments

For direct comparison with other work, datasets are preprocessed exactly as in Dinh et al. (2016). Results are in Figure 11 and Figure 12.

(a) Class conditional CIFAR-10 samples.

(b) Class conditional $32 \times 32$ ImageNet samples.

Figure 10: Class conditional samples on 5-bit CIFAR-10 and $32 \times 32$ ImageNet respectively. Temperature 0.75

Figure 11: Samples from 8-bit, $64 \times 64$ LSUN bedrooms, church and towers respectively. Temperature 1.0.

Figure 12: Samples from an unconditional model with affine coupling layers trained on the CIFAR-10 dataset with temperature 1.0.