[Reviews · NeurIPS 2018]

Reviewer 1



This paper proposes a deep flow based generative model which builds on techniques introduced in the NICE and RealNVP (Dinh 2014,2016). The proposed model has layers of invertible transformations (consisting of 1X1 convolutions and NICE-like affine coupling functions) and some tricks like data dependent activation normalization for stable learning are involved. The empirical results are promising and achieve better log-likelihoods than RealNVP models. Also, the qualitative results are interesting as they result in sharp(non-blurry) images which is unusual for a non-autoregressive or GAN based generative model. The interpolation and face synthesis experiments are especially encouraging. While the results on image generative modeling are promising, the underlying approach itself is not novel and as authors acknowledge, builds upon the work introduced in NICE, RealNVP. My major concern is that I am not able to understand what actually makes the proposed model generate sharp images and better likelihoods. Is it because of the depth (no. of transformations in the flow based model)? Or is it because of the data dependent activation normalization, or is it an effect of some other modeling choice like channel permutation? An ablation based analysis, or even a qualitative analysis of what makes the proposed model work would have been very helpful in understanding the impact of the techniques described in the paper. Thanks to the authors for their response and while I do think that this paper can benefit from a more thorough ablation analysis, I think that good performance of their approach, presumably due to learning the permutation with the new flow, makes this paper a good fit for NIPS.

Reviewer 2



The paper builds on previous work (RealNVP (Dinh et al)) on deep generative models with tractable log-likelihoods. Each layer in RealNVP permutes or reverses the channel dimension and then applies a coupling layer that is invertible. The paper's primary contribution is the addition of a learnable 1x1 invertible convolution to replace fixed permutation matrices used in RealNVP. This applies a linear invertible transformation to the channel dimension and can be seen as a generalisation of permutations. In addition, the paper uses activation normalisation instead of batchnorm. This allows the use of batchsize 1 per GPU and hence allows scaling to larger image sizes. The paper uses gradient checkpointing to scale to move GPUs and deeper architectures. The experimental evaluation is thorough. In a first set of experiments, the authors compare 1x1 to fixed permutations or simply reversing the order. The results show faster convergence and better final performance. Note that 1x1 convolutions linearly blend the different channels, thus allowing different dimensions to affect one another after fewer steps. It could be interesting to study the performance of a fixed 1x1 convolution (say, a random orthogonal matrix) as an additional baseline. Glow outperforms RealNVP in terms of log likelihood on a variety of standard benchmarks. The qualitative section is also very interesting: samples from the models look (subjectively) very realistic, and interpolation in latent space works well. Interestingly, it is possible to vary attributes by perturbing the latent space, using a simple procedure proposed in the paper. Since the paper uses tricks to scale to deeper models than previous work I would be interested in a qualitative comparison with RealNVP. Does Glow beat RealNVP visibly in terms of sample quality and interpolation in the latent space? Is the latent space manipulation proposed for Glow also successful for RealNVP? The paper is very clearly written. While the addition of 1x1 convolutions seems relatively straightforward I believe because of the impressive empirical results both in terms of sample quality and learnt representations this paper will be of big interest to the NIPS community and generate a lot of follow-up work. Thank you to the authors for their response. I continue to think that this paper will be of interest to the NIPS community and look forward to reading the final version.

Reviewer 3



This paper proposes some modifications to Real NVP (Dinh et al.) and claim improved generative modeling performance. Quantitative evaluations (likelihood scores) are used to compare with Dinh et al. on CelebA and Imagenet 32x32 and Imagenet 64x64. The proposed improvements are: 1. Using a learnt invertible channel transformation instead of the fixed shuffling used by Dinh et al. 2. Don't use BatchNorm. Replace it with an affine layer whose parameters are initialized to maintain zero-mean unit-variance. This is called ActNorm in the paper. 3. Generalize Dinh's additive coupling layer into an affine coupling layer. 4. The images are compressed into a 5-bit color space instead of the 8-bit color space typically used. The paper attempts to solve a significant problem (non-autoregressive likelihood-based deep generative models). However, the paper has a number of problems which makes it hard to accept as a NIPS paper. In short, the quantitative and qualitative evaluation fall short of convincing an informed reader about the usefulness of the idea in the paper. The proposed changes might be useful, but as it stands now, this paper doesn't give any evidence. Let me explain in more detail: ## Samples It's not clear how to judge the qualitative samples. The paper only present one set of samples for the same model (the proposed GLOW model) and the same dataset (CelebA 5-bit). To my knowledge, Dinh et al. don't use 5-bit color space and this paper does not provide any samples of any baseline models on the 5-bit color space. So it's simply impossible to properly judge the samples. The paper also doesn't present any nearest-neighbour analysis (in pixel-space or feature-space) to judge overfitting. ## Compute It's not clear whether the contributions are due to the problem-level choices made (such as 5-bit color), solution-level choices (1x1 invertible conv, etc.) or simply more compute used for more iterations. The paper doesn't mention any details about how much compute, gpus or iterations were used in comparison to the Dinh et al. baseline. The authors mention multi-GPU experiments which hints that a relatively large amount of compute was used, and thus doesn't give me confidence that the improvement is due to the solution-level changes. ## Missing Results The paper presents quantitative results for Imagenet datasets, and 5-bit CelebA, but does not provide samples for the same. This indicates the model did not fare well in terms of sample quality in these other harder datasets, which in itself is okay, but skipping out in providing the results is an attempt at misleading readers. Since sample quality is an important claim of the paper, and failing to provide the same for the other datasets (that were trained and evaluated quantitatively) is conspicuous. ## Missing Ablations This is a minor point. The paper makes no attempt at ablating the individual changes proposed. This could have been done on MNIST if compute was an issue. ## Missing comparison to autoregressive models This is a minor point. The paper fails to include likelihood scores for autoregressive models in the table of results. While this doesn't matter for informed readers, it misleads readers who are not tracking this literature into thinking that Glow is a state-of-the-art generative model. # Suggestions for Improvement 1. Compare against the main baseline models (Real NVP) using the same amount of compute. 2. If 1 is not possible, then add a note in the table of results comparing the amount of compute resources used by the two models. 3. Include and discuss samples of the model which were trained on datasets other than CelebA 5-bit. 4. Include nearest-neighbour analysis to judge overfitting. # Summary The paper presents some intuitive modifications to Dinh et al's real NVP model. However, the evaluations fall short of supporting any of the claims made in the paper. ---- # After Author Reponse I had missed figure 3 in my previous read. Figure 3 does provide evidence that the 1x1 convolution improves NLL. I assume this is a controlled setting, i.e. all other variables are the same. Perhaps it would have been useful to mention this in the paper. Since the baseline result is same as the one in the Real NVP paper, it's easy to conclude that the baseline results are picked from the Real NVP paper, and not reproduced in a controlled setting. I will update my review, since this part of the review was based on an oversight. The other parts of my review however still stand. It will be more informative both to reviewers and to readers if the authors also include samples from the controlled baselines (i.e. the reverse and shuffle baselines). I'm sure the authors are well-aware that NLL is not the best metric to compare across different models. Fairly trivial changes in modeling assumptions can hurt or improve NLL scores with pretty much no change in subjective quality of samples. Since we are in the business of evaluating samples subjectively, let's do it somewhat more objectively, by including samples from the baseline. One of the central points in my review was that the authors to change the problem itself by using a 5-bit color space. All the samples shown in the paper are from the 5-bit model. So, while I can judge that the samples "subjectively realistic-looking", it's not clear whether this is due to the choice to shift from 8-bit to 5-bit or the improved modeling. This point remains unaddressed and un-acknowledged by the authors. In summary, I acknowledge that I had missed figure 3's significance in my previous read, and that does improve my judgement of the paper, but the paper still lacks sufficient rigor in evaluation to be accepted as a NIPS paper.